# The BK Channel Limits the Pro-Inflammatory Activity of Macrophages

**DOI:** 10.3390/cells13040322

**Published:** 2024-02-09

**Authors:** Yihe Chen, Nikita Markov, Lea Gigon, Aref Hosseini, Shida Yousefi, Darko Stojkov, Hans-Uwe Simon

**Affiliations:** 1Institute of Pharmacology, University of Bern, 3010 Bern, Switzerland; 2Institute of Biochemistry, Brandenburg Medical School, 16816 Neuruppin, Germany

**Keywords:** AIM2 inflammasome, BK channel, calcium influx, macrophage polarization, pro-inflammatory cytokine, kinase

## Abstract

Macrophages play a crucial role in the innate immune response, serving as key effector cells in the defense against pathogens. Although the role of the large-conductance voltage and calcium-activated potassium channel, also known as the K_Ca_1.1 or BK channel, in regulating neurotransmitter release and smooth muscle contraction is well known, its potential involvement in immune regulation remains unclear. We employed BK-knockout macrophages and noted that the absence of a BK channel promotes the polarization of macrophages towards a pro-inflammatory phenotype known as M1 macrophages. Specifically, the absence of the BK channel resulted in a significant increase in the secretion of the pro-inflammatory cytokine IL-6 and enhanced the activity of extracellular signal-regulated kinases 1 and 2 (Erk1/2 kinases), Ca^2+^/calmodulin-dependent protein kinase II (CaMKII), and the transcription factor ATF-1 within M1 macrophages. Additionally, the lack of the BK channel promoted the activation of the AIM2 inflammasome without affecting the activation of the NLRC4 and NLRP3 inflammasomes. To further investigate the role of the BK channel in regulating AIM2 inflammasome activation, we utilized BK channel inhibitors, such as paxilline and iberiotoxin, along with the BK channel activator NS-11021. Pharmacological inactivation of the BK channel increased, and its stimulation inhibited IL-1β production following AIM2 inflammasome activation in wild-type macrophages. Moreover, wild-type macrophages displayed increased calcium influx when activated with the AIM2 inflammasome, whereas BK-knockout macrophages did not due to the impaired extracellular calcium influx upon activation. Furthermore, under conditions of a calcium-free medium, IL-1β production following AIM2 inflammasome activation was increased in both wild-type and BK-knockout macrophages. This suggests that the BK channel is required for the influx of extracellular calcium in macrophages, thus limiting AIM2 inflammasome activation. In summary, our study reveals a regulatory role of the BK channel in macrophages under inflammatory conditions.

## 1. Introduction

Monocyte-derived macrophages and tissue-resident macrophages together constitute the natural macrophage system, serving as the host’s first line of defense [1,2]. Under physiological conditions, macrophages play a routine role in immune surveillance and in maintaining tissue homeostasis [3]. They are essential as key effector cells in innate immune responses and also communicate with cells of the adaptive immune system [4]. In the initial stages of inflammation, macrophages undergo a transition to a pro-inflammatory state known as the M1 macrophage phenotype. This phenotype is characterized by the production of nitric oxide (NO), reactive oxygen species (ROS), and the release of pro-inflammatory cytokines, such as IL-6 and TNF-α [5]. In contrast, M2 macrophages, which are associated with the resolution phase of inflammation [6], exhibit an anti-inflammatory phenotype. M2 macrophages express arginase 1 and the anti-inflammatory cytokine IL-10 [7]. There is evidence that the pharmacological modulation of the macrophage phenotype is an effective strategy for treating a variety of inflammatory diseases and cancer [8].

Macrophages are known to express a wide variety of ion channels to move ions across their cell membranes. The flow of ions is crucial for maintaining essential macrophage characteristics and functions [9]. Among these ion channels, potassium (K^+^) channels are particularly important as they regulate the cell membrane potential and play a critical role in modifying cell functions through their interaction with calcium (Ca^2+^) channels [10]. One such K^+^ channel is the large-conductance voltage and Ca^2+^-activated K^+^ channel, also known as MaxiK, K_Ca_1.1, Slo1, or BK channel characterized by its high conductance of 100–300 pico Siemens (pS) [11]. The BK channel is a tetramer of four identical α-subunits encoded by the *Kcnma1* gene. Each α-subunit contains three structural domains with distinct functions: a pore-gate domain (PGD), a voltage sensor domain (VSD), and a large cytosolic tail domain (CTD) [12]. Under physiological conditions, the pores of the BK channel, which control the flow of K^+^, are normally closed. However, they can be activated via membrane depolarization and/or increased levels of Ca^2+^ inside the cell [13]. The BK channel is well known for its role in the regulation of neurotransmitter release [14,15,16] and smooth muscle contractility [17,18]. The BK channel has also been implicated in the regulation of tumor cell proliferation [19,20,21], gene expression [22], airway inflammation [23,24], hormone secretion [25,26], and cell volume [27]. Furthermore, it is considered a promising therapeutic target for hearing loss [28]. Currently, the BK channel is considered to have either no or little roles in the function of immune cells [29]. Notably, T and B lymphocytes do not express the BK channel [10,30,31,32,33], and its relevance in neutrophils [34,35] and eosinophils [36] remains unclear. On the other hand, it has been reported that the BK channel may modulate macrophage phagocytosis [37]. However, a deep understanding of the BK channel’s role in regulating macrophage behavior and function does not currently exist [38].

An inflammatory response is triggered in reaction to infection and tissue injury [39]. Macrophages, being pivotal multifunctional cells, play a critical role in both initiating and resolving inflammation. They effectively eliminate invading pathogens and dead cells via phagocytosis [40]. Macrophages detect pathogenic microorganisms by expressing pattern-recognition receptors (PRRs), which enable them to identify non-self-objects expressing pathogen-associated molecular patterns (PAMPs). PRRs can also be activated via damage-associated molecular patterns (DAMPs) released from damaged or dead cells [41,42]. PRRs fall into five structural categories: Toll-like receptors (TLRs), nucleotide oligomerization domain (NOD)-like receptors (NLRs), absent in melanoma-2 (AIM2)-like receptors (ALRs), retinoic acid-inducible gene-I (RIG-I)-like receptors (RLRs), and C-type lectin receptors (CLRs) [43]. Specific NLRs and ALRs intracellularly recognize PAMPs and DAMPs, initiating inflammasome assembly [44]. Among these inflammasomes, NLRP3, NLRC4, and AIM2 have been extensively studied. Inflammasome activation typically involves the structural interaction of sensor proteins with the adaptor protein ASC (the adaptor molecule apoptosis-associated speck-like protein containing a CARD) and casapase-1, resulting in complex oligomerization, the autocatalytic cleavage of caspase-1, and the maturation of IL-1β and IL-18 [45,46]. The activation of inflammasomes is a crucial mechanism in converting danger signals caused by infection and cellular damage into an inflammatory response. This process plays a significant role in the development of various inflammatory disorders [47].

In this study, we provide compelling evidence for the role of the BK channel in macrophages in the regulation of their pro-inflammatory responses. The absence of the BK channel in macrophages results in increased M1 macrophage polarization, IL-6 secretion, and AIM2 inflammasome activation. Furthermore, our investigation has revealed the involvement of Erk1/2 and CaMKII kinases, as well as the ATF-1 transcription factor, which are associated with BK channel activity.

## 2. Materials and Methods

### 2.1. Reagents and Materials

Iscove’s modified Dulbecco’s medium (IMDM), lipopolysaccharide (LPS), Luria broth base (LB) medium, nigericin, Tris_Base, sodium chloride (NaCl), glycerol, sodium pyrophosphate, sodium orthovanadate (Na_3_VO_4_), phenylmethylsulfonyl fluoride (PMSF), protease inhibitor cocktail, Tween 20, potassium chloride (KCl) solution, sodium chloride (NaCl) solution, magnesium chloride (MgCl_2_) solution, HEPES solution, dithiothreitol (DTT) solution, Dulbecco’s phosphate buffered saline (PBS), EDTA, sodium fluoride (NaF), and acetone were purchased from Sigma-Aldrich (Buchs, Switzerland). Fetal calf serum (FCS) was obtained from GE Healthcare Life Sciences (Little Chalfont, UK). Interferon gamma (IFN-γ) and interleukin-4 (IL-4) were obtained from Peprotech (London, UK). Ultrapure LPS, adenosine triphosphate (ATP), poly(dA:dT), and ultrapure flagellin were purchased from Invivogen (San Diego, CA, USA). Lipofectamine 2000 transfection reagent and Fluo-4 were supplied from Invitrogen (Carlsbad, CA, USA). The DOTAP liposomal transfection reagent, PhosSTOP phosphatase inhibitor cocktail, and dithiothreitol (DTT) were purchased from Roche Diagnostics (Rotkreuz, Switzerland). Triton X-100 and PVDF membrane were purchased from Merck Millipore (Darmstadt, Germany). Bovine serum albumin (BSA) was obtained from PAN-Biotech (Aidenbach, Germany). The calcium chloride (CaCl_2_) solution was purchased from Amresco (Solon, OH, USA). The RNA lysis buffer and RNA MicroPrep kit were obtained from Zymo Research (Irvine, CA, USA). Random primers, dNTP mix, RNasin plus RNase inhibitor, Griess reagent system kit, and CellTiter-Glo luminescent cell viability assay kit were supplied by Promega AG (Dübendorf, Switzerland). The iTaq Universal SYBR Green supermix was from Bio-Rad Laboratories AG (Cressier, Switzerland). Agarose tablets were from Bioline (London, UK). The RedSafe nucleic acid staining solution was from iNtRON Biotechnology (Seongnam, Republic of Korea). Normal rat serum (NRS) was from Dako, distributed by Agilent Technologies Schweiz AG (Basel, Switzerland). Normal Syrian hamster serum was from Jackson ImmunoResearch (West Grove, PA, USA). GlutaMAX supplement, penicillin-streptomycin, 2-mercaptoethanol, Opti-MEM reduced serum medium, Pierce BCA protein assay kits, SuperScript III reverse transcriptase Kit, Disuccinimidyl suberate (DSS), and Hank’s Balanced Salt Solution (HBSS) were obtained from Thermo Fisher Scientific (distributed by LuBioScience, Lucerne, Switzerland). Paxilline and iberiotoxin were supplied from Tocris Bioscience (Bristol, UK). NS-11021 was from Alomone Labs (Jerusalem, Israel). Black, glass-bottom 24-well plates and opaque-walled white 96-well plates were purchased from Greiner Bio-One (Frickenhausen, Germany).

### 2.2. Mice

Mice were kept in specific pathogen-free facilities in individually ventilated cages (Tecniplast, Green Line, Buguggiate, Italy), with a 12 h light/12 h dark cycle and autoclaved acidified water and cages, including food, bedding, and environmental enrichment. All experiments were performed with age- and sex-matched mouse cohorts. Mice were on a *SV129-C57BL/6* genetic background. *BK*^−/−^ mice were kindly provided by Dr. Peter Ruth (University of Tübingen, Department of Pharmacology, Toxicology and Clinical Pharmacy, Institute of Pharmacy, Tübingen, Germany). *BK*^−/−^ mice were generated via the depletion of the BK channel α-subunit encoded by the *kcnma1* gene [48]. F1 generation *BK*^−/−^ mice were bred by mating *SV129 BK*^+/−^ and *C57BL/6 BK*^+/−^ mice and identified using the genotyping method. Bone marrow cells were isolated from 6–8-week-old WT and *BK^−/−^* mice. Approval for all mouse experiments was obtained from the Cantonal Veterinary Office of Bern, Switzerland, and all experiments were conducted in accordance with Swiss federal legislation on animal welfare.

### 2.3. Cell Culture

Femurs and tibias were obtained from WT and *BK^−/−^* mice. Briefly, bone marrow cells were collected by flushing the femurs and tibias with an isolation medium (PBS with 2% FCS) with a 26-gauge needle and filtering through a sterile, 70 μm nylon cell strainer (Greiner Bio-One, Leipzig, Germany). The procedure was performed on ice. The cell number was counted with a hematology analyzer (Sysmex Suisse AG, Yverdon-les-Bains, Switzerland). For the differentiation of the bone marrow-derived macrophages (BMDMs), 5.5 × 10^6^ bone marrow cells were cultured in a 75 cm^2^ flask (Greiner Bio-one) in a medium containing 70% IMDM (10% FCS, 1% Glutamine, and 1% Penicillin/Streptomycin), 30% L929 conditioned medium, and 0.05% 2-Mercaptoethanol (50 mM) (Gibco). After 8 days of differentiation, BMDMs were collected for experimental analysis.

### 2.4. Macrophage Polarization

BMDMs were polarized to pro-inflammatory (M1) macrophage with 100 ng/mL LPS (Sigma-Aldrich) and 20 nM IFN-γ (Peprotech). Anti-inflammatory (M2) macrophage polarization was achieved by treating BMDMs with 20 nM IL-4 (Peprotech). Nontreated BMDMs were cultured in a parallel manner in IMDM supplemented with a 10% L929 conditioned medium as the control and abbreviated as M0 macrophages. A total of 10% L929 conditioned medium was supplemented with IMDM for M1 and M2 macrophage polarization when compared with M0 macrophages.

### 2.5. Nitrite Production Measurement

BMDMs were plated in a 6-well plate, 0.65 × 10^6^ cell/well, and polarized to M1 macrophages for 24 h. The cell culture supernatant was collected. Nitrite concentration was quantified using a Griess Reagent System assay kit (Promega) as described in the manual and measured with a spectrofluorometer (SpectraMax M2, Molecular Devices, San Jose, CA, USA).

### 2.6. Phagocytosis Assay

One single colony of GFP-expressing *Escherichia coli* (GFP-*E. coli*) (strain MG1655, a kind gift of Dr. Emma Slack, ETH Zurich) was cultured in a Luria broth base (LB) medium (Sigma-Aldrich) overnight with shaking at 220 rpm, 37 °C. The bacterial culture was diluted 1:100 in a LB medium, grown to a mid-logarithmic growth phase (OD600 = 0.7), and centrifuged at 1000× *g* for 5 min. The bacteria were washed twice with 1× HBSS and then gently centrifuged at 100× *g* for 5 min to remove clumped bacteria. The 0.25 × 10^6^ cell/well BMDMs were plated on a 12-well plate (Corning) and polarized to M1 and M2 macrophages for 24 h. M0 macrophages were cultured parallelly and used as control cells. Cells were washed with PBS, and IMDM without antibiotics was added to the well. The 2.5 × 10^6^ GFP-*E.coli* was added to each well and incubated at 37 °C for 30 min. Cells were washed three times with PBS to remove non-internalized bacteria. Cells were collected and analyzed with a BD FACSLyric flow cytometer (BD Biosciences, Allschwil, Switzerland).

### 2.7. Inflammasome Activation

BMDMs were plated in a 6-well plate (0.65 × 10^6^ cell/well) or a 24-well plate (0.13 × 10^6^ cell/well) based on the experimental assignment. Inflammasome activation was performed in two steps. BMDMs were primed with 100 ng/mL ultrapure LPS (Invivogen) in an IMDM containing 10% FCS, 1% Glutamine, and 1% Penicillin/Streptomycin for 4 h, followed by inflammasome activation. For NLRP3 inflammasome activation, LPS-primed BMDMs were treated with 20 µM nigericin (Sigma-Aldrich) for 30 min or 5 mM ATP (Invivogen) for 45 min. For AIM2 inflammasome activation, 1 µg/mL Poly(dA:dT) (Invivogen) was mixed with 2.5 µL/mL lipofectamine 2000 (Invitrogen) in Opti-MEM (Gibco) and incubated for 10 min at room temperature (RT). Cells were treated with Poly(dA:dT) complexes for 50 min to activate the AIM2 inflammasome. For NLRC4 inflammasome activation, 0.5 µg/mL ultrapure flagellin (Invivogen) was mixed with 25 µL/mL DOTAP (Roche) in IMDM and incubated for 10 min at RT. Cells were treated with flagellin complexes for 1 h to activate the NLRC4 inflammasome.

### 2.8. ELISA

For IL-1β analysis, BMDMs were plated in a 24-well plate (0.13 × 10^6^ cell/well), and inflammasomes were activated after LPS priming for 4 h. For IL-6 and TNF-α analysis, BMDMs were plated in a 6-well plate (0.65 × 10^6^ cell/well) and polarized to M1 macrophages for 24 h in IMDM. The cell culture supernatant was collected and centrifuged at 1500 rpm, 4 °C for 5 min to remove cell debris. Samples were stocked at −80 °C. For cytokine analysis, samples were centrifuged at 13,000 rpm, 4 °C for 10 min. The supernatant was collected, and IL-1β was measured with the ELISA MAX™ Deluxe Set Mouse IL-1β assay kit (Biolegend, San Diego, CA, USA). Samples were diluted for the measurement of IL-6 (1:100) and TNF-α (1:20) and analyzed with the ELISA MAX™ Deluxe Set Mouse IL-6 and ELISA MAX™ Deluxe Set Mouse TNF-α assay kit (Biolegend) according to the manufacturers’ instructions.

### 2.9. Immunoblotting

Cell lysates were prepared by lysing the cells with the lysis buffer containing 50 mM Tris (pH 7.4), 150 mM NaCl, 10% Glycerol, 1% Triton X-100, 2 mM EDTA, 10 mM NaPyrophosphate, 50 mM NaF, and 200 μM Na_3_VO_4_, supplemented with 1 mM PMSF, protease inhibitor cocktail (Sigma-Aldrich) and 1×PhosSTOP phosphatase inhibitor cocktail (Roche). Cells were collected, lysed, and incubated on ice for 30 min. Cell lysates were collected via centrifugation at 13,000 rpm for 15 min at 4 °C, and protein concentrations were quantified with a Pierce BCA protein assay kit (Thermo Fisher Scientific). Protein samples were prepared with a DTT (Roche) and 4x Laemmli protein sample buffer (Bio-Rad). Protein samples (45 μg) were heated at 95 °C for 5 min and loaded into a SERVAGel TG PRiME gel (SERVA Electrophoresis, Heidelberg, Germany) and transferred to a PVDF membrane (Immobilion-P; Merck Millipore) after electrophoresis. The membrane was blocked with 5% nonfat milk or 5% BSA (PAN-Biotech) in TBST (0.1% Tween 20 in 20 mM Tris and 150 mM NaCl [pH 7.6]) for 1 h and incubated overnight at 4 °C with primary antibodies: rabbit monoclonal anti-iNOS (1:1000, Cell Signaling Technology, Danvers, MA, USA), rabbit monoclonal anti-Phospho-p44/42 MAPK (Erk1/2) (Thr202/Tyr204) (1:1000, Cell Signaling Technology), rabbit monoclonal anti-p44/42 MAPK (Erk1/2) (1:1000, Cell Signaling Technology), rabbit monoclonal anti-Phospho-CaMKII (Thr286) (1:500, Cell Signaling Technology), rabbit polyclonal anti-CaMKII (pan) (1:1000, Cell Signaling Technology), rabbit monoclonal anti-Phospho-CREB (Ser133) (1:1000, Cell Signaling Technology), rabbit monoclonal anti-Phospho-NF-κB p65 (Ser536) (1:1000, Cell Signaling Technology), rabbit polyclonal anti-Phospho-Stat1 (Tyr701) (1:1000, Cell Signaling Technology), rabbit polyclonal anti-Stat1 (1:1000, Cell Signaling Technology), rabbit polyclonal anti-NFκB p65 (1:1000, Santa Cruz Biotech, Heidelberg, Germany), mouse monoclonal anti-Caspase-1 (p20) (1:1000, Adipogen, Fuellinsdorf, Switzerland), rabbit polyclonal anti-Asc (1:1000, Adipogen), mouse polyclonal anti-IL-1β (1:1000, RD Systems, Minneapolis, MN, USA), rabbit polyclonal anti-Pan Actin (1:10,000, Cytoskeleton, Denver, CO, USA), and rabbit polyclonal anti-β-Actin (1:1000, Cell Signaling Technology). The membrane was washed with TBST three times, each 10 min, and incubated with the corresponding HRP-conjugated secondary antibody: anti-mouse IgG (Amersham), anti-rabbit IgG (Amersham), or anti-goat IgG (Dako) in 1:5000 dilution for 1 h at RT. After washing with TBST three times, each 10 min, the protein signal was detected with an LI-COR Odyssey imaging system and analyzed using Image Studio software (latest v. 5.5 LI-COR Biosciences, Lincoln, NE, USA).

### 2.10. Supernatant Protein Precipitation

BMDMs (0.65 × 10^6^ cell/well) were plated in a 6-well plate and primed with LPS for 4 h followed by inflammasome activation. The supernatant was collected via centrifugation at 13,000 rpm, 4 °C for 10 min. The supernatant was aliquoted 300 μL/tube for protein precipitation. The sample volume of cold acetone was added to the sample four times. Incubate at −20 °C for 1 h. Samples were centrifuged at 4 °C, 13,000 rpm for 10 min. The supernatant was disposed of, and the cell pellet was washed with 0.5 mL cold acetone. Cells were again centrifuged, and the remaining acetone was discarded. The cell pellet was dissolved in water, and protein samples were prepared with a DTT and 4×Laemmli protein sample buffer. Samples were heated at 95 °C for 5 min and resolved via SDS-PAGE. An immunoblot assay was performed to evaluate the level of cleaved IL-1β and Caspase-1 (p20) in the cell culture supernatant.

### 2.11. ASC Oligomerization

BMDMs (0.65 × 10^6^ cells/well) were plated in a 6-well plate, and inflammasomes were activated. Cell lysates were collected via centrifugation at 13,000 rpm, 4 °C for 15 min. The lysate pellet was washed twice with ice-cold PBS containing proteinase inhibitors and resuspended in PBS containing 2 mM DSS, incubated for 30 min at RT. The supernatant was removed, and the DSS-treated pellet was lysed with a lysis buffer. Samples were prepared with a DTT and 4×Laemmli protein sample buffer and analyzed with an immunoblot assay.

### 2.12. Solution

Ca^2+^ free solution: 5 mM KCl, 145 mM NaCl, 1 mM MgCl_2_, and 10 mM HEPES; 2 mM Ca^2+^ solution: 2 mM CaCl_2_, 5 mM KCl, 145 mM NaCl, 1 mM MgCl_2_, and 10 mM HEPES [49].

### 2.13. Quantitative RT-PCR

Cells were washed with cold PBS and lysed in an RNA Lysis Buffer (Zymo Research) containing a DTT (Sigma-Aldrich) and RNase inhibitor (Promega AG). Total RNA was extracted using the RNA MicroPrep kit (Zymo Research) following the manufacturer’s instructions. cDNA synthesis was performed using 1000 ng RNA, in the presence of random primers, dNTP mix, and RNasin Plus RNase inhibitor (all from Promega AG) using the SuperScript III Reverse Transcriptase Kit (Thermo Fisher Scientific) and following the three-step program (25 °C 5 min, 50 °C 1 h, and 70 °C 15 min). Real-time PCR was performed using the iTaq Universal SYBR Green Supermix (Bio-Rad Laboratories) with the CFX Connect Real-Time PCR Detection system (Bio-Rad Laboratories). The primers that were used in this study are listed in Table 1. Primers were synthesized using Microsynth AG, (Balgach, Switzerland). The data were analyzed using the comparative cycle-threshold (CT) method with the reference gene *Eps8l1*. The relative expression of the targeted gene was calculated as normalized to the control (wild type).

The RT-PCR products of *Kcnma1* (BKα) were separated with 4% agarose gel (Bioline) stained with red safe (iNtRON Biotechnology), run at 120 V, and visualized with the Gel Doc 2000 gel documentation system (Bio-Rad Lab. AG, Reinach, Switzerland). *Actb* was used as a reference.

### 2.14. Flow Cytometry

BMDMs were plated in a 12-well plate (Corning), 0.25 × 10^6^ cell/well, and polarized to M1 macrophages in IMDM. Nontreated BMDMs were cultured as M0 macrophages. Cells were washed with PBS and collected via centrifugation at 250× *g*, 4 °C for 5 min. Cells were washed with a washing buffer (PBS with 2% FCS) and centrifuged at 350× *g*, 4 °C for 5 min. The supernatant was discarded, and the Fc receptors were blocked by 10 min incubation on ice in a blocking buffer (PBS with 2% FCS, 20% 2.4G2 supernatant, 3% NRS, and 5% normal Syrian hamster serum). Cells were incubated with fluorochrome-labeled monoclonal antibodies in a blocking buffer for 20 min on ice. Anti-CD11b PE-Cyanine7 (eBioscience, San Diego, CA, USA) and anti-CD86 APC-R700 (BD Biosciences) antibodies were used for flow cytometry analysis. Cells were washed with a washing buffer and fixed in 2% paraformaldehyde (PFA) at RT for 10 min. Samples were measured with a BD FACSLyric flow cytometer (BD Biosciences), and data were subsequently analyzed with FlowJo 7.6.5. software (Tree Star, Ashland, OR, USA).

### 2.15. ATP Measurement

BMDMs were plated in an opaque-walled white 96-well plate (Greiner Bio-One), 25,000 cell/well. M0, M1, and M2 macrophages were polarized and cultured for 24 h for ATP analysis. The amount of ATP was quantified using the CellTiter-Glo luminescent cell viability assay kit (Promega) according to the manufacturer’s instructions. The luminescent signal was recorded using the GloMax Explorer multimode microplate reader (Promega AG, Dübendorf, Switzerland).

### 2.16. Calcium Imaging Assay

BMDMs were plated in a black, glass-bottom 24-well plate (Greiner Bio-One), 0.25 × 10^6^ cell/well. Cells were incubated with a cell-permeable Ca^2+^-indicator Fluo-4 AM (Invitrogen) in Opti-MEM for 30 min at 37 °C. Cells were primed with LPS for 4 h, followed by AIM2 inflammasome activation. Ca^2+^ mobilization was recorded in live cell microscopy experiments using confocal laser scanning microscopy (LSM 800, Carl Zeiss Micro Imaging GmbH, Jena, Germany) with a Plan-Aprochrom at a 40×/1.4 Oil DIC objective. For each experiment, we selected three representative positions with at least 20 cells per condition and subsequently obtained pictures at different time points (baseline, priming, and activation). The mean fluorescence intensity (MFI) of Fluo-4 AM (green channel) of single cells representing calcium mobilization was analyzed for each image separately using the automated surface module of Imaris software (latest v. 10.1 Bitplane AG, Zurich, Switzerland).

### 2.17. Statistical Analysis

The data were analyzed using an unpaired Student *t*-test or two-way ANOVA followed by pairwise comparisons using Sidak’s or Tukey’s correction using the GraphPad Prism 6 software (San Diego, CA, USA) and presented as means ± SD. *P*-values of <0.05 were considered statistically significant.

## 3. Results

### 3.1. Lack of BK Channel Enhances the Pro-Inflammatory Activities of M1 Macrophages

To explore the influence of the BK channel in regulating macrophage functions, including macrophage polarization, release of pro-inflammatory mediators, phagocytosis, and metabolism, we generated BMDMs representing M0 macrophages from wild-type and BK-knockout mice (Appendix A). The knockout effect of the Kcnma1 gene, which encodes the α-subunit of the BK channel, was confirmed using a RT-PCR assay (Figure 1A). M0 macrophages were induced to polarize into M1 and M2 phenotypes. We observed an increase in the expression of iNOS, a hallmark of M1 macrophages, in the BK-knockout macrophages (Figure 1B), along with increased nitrite production (Figure 1C). The absence of the BK channel resulted in an upregulation of mRNA levels for M1 macrophage markers, Nos2, Il6, Il1b, and Tnf (Figure 1D) while simultaneously reducing the mRNA levels of the M2 macrophage markers Arg1 and Chi3l3 (Appendix A). Macrophage phagocytosis was not affected in BK-knockout macrophages (Appendix A). Furthermore, adenosine triphosphate (ATP) production, which governs cellular respiration activity, exhibited no significant difference between wild-type and BK-knockout macrophages (Appendix A). These results indicate that the absence of the BK channel in macrophages may support the development of a pro-inflammatory phenotype (M1). We next explored the absence of the BK channel on additional characteristics of M1 macrophages. The secretion of pro-inflammatory cytokines by M1 macrophages constitutes a critical step in mediating macrophage-driven inflammation. Notably, the absence of the BK channel resulted in increased production and release of IL-6 (Figure 1E) but had no significant impact on TNF-α release (Figure 1F). Macrophages exhibit diverse cell surface markers, which allow for categorizing distinct phenotypes. We utilized the widely recognized macrophage surface marker CD11b to differentiate between macrophage populations [50]. CD86 is a hallmark of M1 macrophage polarization. In evaluating the disparity in M1 between wild-type and BK-knockout macrophages, we analyzed the expression of CD86 in M0 and M1 macrophages. We observed no significant difference in the expression level of CD86 between wild-type and BK-knockout macrophages (Appendix A). In summary, the data presented provide conclusive evidence that the absence of the BK channel fosters the polarization of M1 macrophages while impeding the polarization of M2 macrophages. Furthermore, our findings underscore the crucial role of the BK channel in governing the production of the pro-inflammatory cytokine IL-6.

### 3.2. Lack of BK Channel Results in Increased Activity of Kinases Erk1/2, CaMKII, and the Transcription Factor ATF-1

The BK channel is located in the cell membrane and primarily functions to enable the efflux of K^+^ from the cell interior into the extracellular space upon activation. Opening the BK channel allows for a quick flow of K^+^ out of the cell, leading to a significant change in the cell membrane potential and the initiation of a cascade of cell signaling processes that trigger various cellular events [51]. To delve deeper into the impact of the BK channel on M1 macrophage polarization, we explored possible molecular mechanisms. Cellular signaling pathways, comprising various kinases and their subsequent targets, hold a pivotal role in governing cellular behavior. Interestingly, we observed heightened Erk1/2 activity in BK-knockout M1 macrophages (Figure 2A). Previous studies have indicated that CaMKII can regulate Erk1/2 activation in RAW264.7 macrophages [52]. Indeed, during M1 macrophage polarization, we also observed an increase in CaMKII phosphorylation in BK-knockout compared to wild-type macrophages (Figure 2B).

CREB is capable of regulating IL-6 expression [53], while ATF-1 acts as its co-activator to regulate target gene expression [54,55]. We observed increased phosphorylation of the transcription factors CREB/ATF-1 in BK-knockout compared to wild-type M1 macrophages, with a more pronounced effect on ATF-1 (Figure 2C). NF-κB and STAT1 are well-known transcription factors that mediate M1 macrophage polarization [56]. However, the activities of NF-κB (p65) and STAT1 displayed no significant differences between wild-type and BK-knockout macrophages (Figure 2C). Taken together, these findings suggest that the absence of the BK channel leads to increased activity of kinases Erk1/2, CaMKII, and the transcription factor ATF-1 in M1 macrophages.

### 3.3. Lack of BK Channel or Its Pharmacological Inactivation Results in Increased IL-1β Production: Evidence for AIM2, but Not NLRC4 and NLRP3 Inflammasome Regulation

To explore the influence of the BK channel on inflammasome activation, BMDMs were initially primed with LPS for 4 h, followed by specific stimuli to induce AIM2, NLRC4, or NLRP3 inflammasome activation. Assessment of inflammasome activation included measurements of IL-1β release, caspase-1 cleavage, and ASC oligomerization. In the absence of the BK channel, an increase in IL-1β release was observed upon AIM2 inflammasome activation. However, no discernible difference was noted in the release of IL-1β induced via NLRC4 or NLRP3 inflammasome activation between wild-type and BK-knockout macrophages (Figure 3A). To further substantiate the role of the BK channel in AIM2 inflammasome activation, immunoblot analysis was conducted. Following activation, BK-knockout macrophages exhibited elevated levels of mature IL-1β and caspase-1 (p20) in supernatants, alongside ASC oligomerization in the cell pellet, thus confirming the regulatory role of the BK channel in AIM2 inflammasome activation (Figure 3B). In contrast, the absence of the BK channel did not impact the activation of NLRC4 or NLRP3 inflammasomes (Figure 3B).

To further analyze the specific involvement of the BK channel in AIM2 inflammasome activation, we employed paxilline and iberiotoxin (IbTX), two recognized pharmacological inhibitors of the BK channel, as well as the BK channel activator NS-11021. In wild-type macrophages, both paxilline and IbTX led to increased IL-1β release, while they had no effect in BK-knockout macrophages (Figure 3C). Additionally, NS-11021 inhibited AIM2 inflammasome activation in wild-type but not BK-knockout macrophages (Figure 3D).

### 3.4. The BK Channel Controls AIM2 Inflammasome Activation by Regulating the Influx of Extracellular Ca^2+^

The BK channel plays a crucial role in maintaining intracellular calcium ([Ca^2+^]_i_) homeostasis [57]. Hence, we investigated whether the BK channel regulates AIM2 inflammasome activation by modulating [Ca^2+^]_i_ levels. To monitor [Ca^2+^]_i_ levels during AIM2 inflammasome activation, we utilized Fluo-4 fluorescence to track and record the Ca^2+^ signals at various stages, including the resting state, LPS priming, and AIM2 inflammasome activation, using confocal microscopy. We observed no difference in [Ca^2+^]_i_ levels between wild-type and BK-knockout macrophages in the resting state and following LPS priming (Figure 4A). However, upon AIM2 inflammasome activation, [Ca^2+^]_i_ concentrations dramatically increased in wild-type but remained almost unaltered in BK-knockout macrophages (Figure 4A). These findings point to the possibility that the activation of the AIM2 inflammasome relies on [Ca^2+^]_i_.

To further investigate the impact of Ca^2+^ influx on AIM2 inflammasome activation, we utilized two distinct media: a Ca^2+^-free solution and a control solution containing 2 mM Ca^2+^. Intriguingly, under Ca^2+^-free conditions, there was a significant increase in IL-1β release observed in both wild-type and BK-knockout macrophages, suggesting that the absence of extracellular Ca^2+^ influx promotes AIM2 inflammasome activation (Figure 4B). These findings imply that the absence of the BK channel restricts extracellular Ca^2+^ influx in macrophages, ultimately augmenting AIM2 inflammasome activation.

## 4. Discussion

Macrophages display diverse phenotypes in response to microenvironmental stimuli. They can polarize towards pro-inflammatory (M1) or anti-inflammatory (M2) phenotypes, which exhibit distinct gene expression, signaling pathways, and functions that are crucial in various diseases [58,59]. There has been a significant interest in regulating macrophage polarization, especially in the development of new therapies for inflammatory diseases and cancer. In our study, we demonstrate the regulatory role of the BK channel, which can limit the pro-inflammatory activity of macrophages under certain conditions. Therefore, pharmacological activation of the BK channel might be beneficial in chronic inflammatory diseases, while its inactivation could represent a new strategy to treat cancer.

M1 macrophages are known to produce high amounts of pro-inflammatory cytokines upon activation, especially IL-6, TNF-α, and IL-1β [60]. Our findings have highlighted a significant increase in IL-6 production in the absence of BK channel expression. Moreover, the lack of BK channel expression intensified the release of IL-1β by enhancing the activation of the AIM2 inflammasome. Additionally, BK-knockout M1 macrophages exhibited increased Erk1/2 activity, as well as elevated CaMKII and increased ATF-1 phosphorylation, suggesting potential mechanisms regulating M1 macrophage functions.

The AIM2 inflammasome critically initiates immune responses linked to specific infections and cellular injuries [61,62,63]. Various mechanisms inhibit AIM2 inflammasome activation, including competitive interactions between components of inflammasome and ligand DNA, as well as reduced protein levels of AIM2 [46,64,65]. We demonstrate that the lack of a BK channel specifically promoted AIM2 inflammasome activation without affecting NLRC4 and NLRP3 inflammasomes. Furthermore, the BK channel inhibitors paxilline and iberiotoxin enhanced AIM2 inflammasome activation, while the BK channel activator NS11021 suppressed its activation. Our findings strongly support the role of the BK channel as a negative regulator of AIM2 inflammasome activation.

The activation of the BK channel leads to a swift efflux of K^+^ ions that hyperpolarize the membrane potential [66]. It has been previously reported that voltage-dependent Ca^2+^ channels are closed to prevent excessive Ca^2+^ influx, which is harmful to cells [67]. It is also known that the BK channel interacts with Ca^2+^ channels to control Ca^2+^ influx [68]. In this report, we demonstrate that during AIM2 inflammasome activation, wild-type macrophages exhibited increased Ca^2+^ levels, while BK-knockout macrophages lacked this response. Blocking the influx of extracellular Ca^2+^ augmented AIM2 inflammasome activation in both wild-type and BK-knockout macrophages, indicating that increases in intracellular Ca^2+^ limit the activation of the AIM2 inflammasome. In summary, our results suggest that the absence of the BK channel hinders Ca^2+^ influx, promoting AIM2 inflammasome activation in macrophages. The Ca^2+^ channel interacting with the BK channel in macrophages remains to be identified.

## 5. Conclusions

In this study, we have uncovered the pivotal role of the BK channel in modulating the pro-inflammatory functions of macrophages (Figure 5). The absence of the BK channel promotes M1 but inhibits M2 macrophage polarization. Notably, the absence of the BK channel amplifies the production of the pro-inflammatory cytokine IL-6. Pharmacological inactivation of the BK channel increases IL-1β release by augmenting AIM2 inflammasome activation. Therefore, the BK channel appears to be a negative regulator of pro-inflammatory activities of macrophages.

## Figures and Tables

**Figure 1 cells-13-00322-f001:**
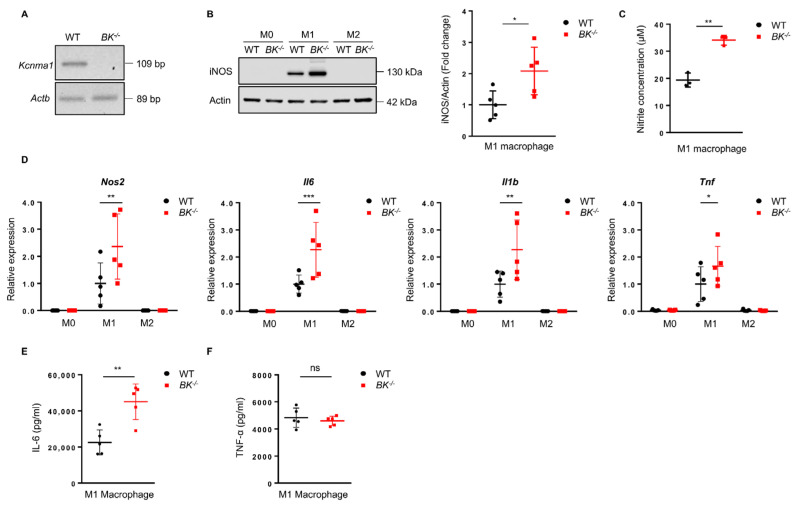
The absence of the BK channel in macrophages supports M1 polarization and pro−inflammatory cytokine release. (**A**) Analysis of BK mRNA expression in wild−type and BK−knockout macrophages was conducted using RT−PCR and visualized via agarose gel electrophoresis. (**B**) Immunoblot analysis of iNOS expression was performed 24 h after initiation of macrophage polarization (left). Quantification of iNOS level in M1 macrophages was normalized to the actin loading control (right). Values are means ± SD. * *p* < 0.05. All data are representative of five independent experiments. (**C**) Quantification of nitrite production in the cell-cultured supernatant was conducted 24 h after initiation of M1 macrophage polarization. Values are means ± SD. ** *p* < 0.01. Each graph represents the results of three independent experiments. (**D**) Gene expression analysis of *Nos2*, *Il*−6, *Il*−1β, and *Tnf* in M0, M1, and M2 macrophages using qPCR. Values are means ± SD. * *p* < 0.05, ** *p* < 0.01, *** *p* < 0.001. Each graph represents the results of five independent experiments. (**E**) ELISA analysis was performed to measure IL−6 release 24 h after initiation of M1 macrophage polarization. Values are means ± SD. ** *p* < 0.01. Each graph represents the results of five independent experiments. (**F**) TNF-−α release was assessed using ELISA 24 h after initiation of M1 macrophage polarization. Values are means ± SD. ns, not significant. Each graph represents the results of five independent experiments.

**Figure 2 cells-13-00322-f002:**
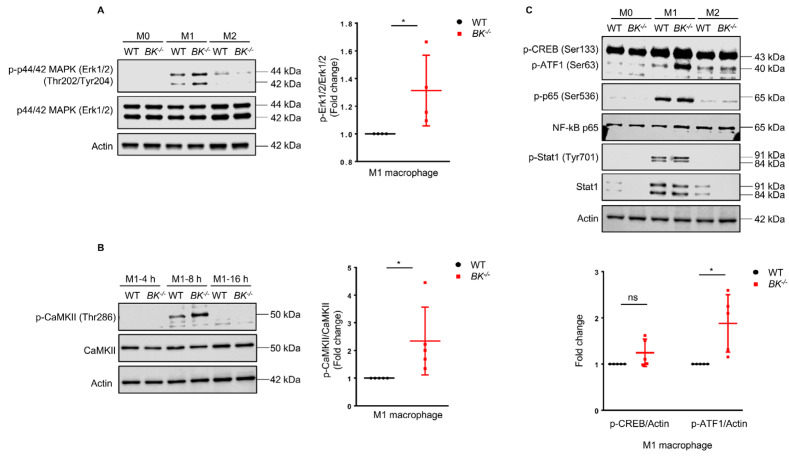
The absence of the BK channel increases the phosphorylation of Erk1/2, CaMKII, and ATF-1 in M1 macrophages. (**A**) BMDMs were polarized into M1 and M2 macrophages for 24 h, while untreated M0 macrophages were used as controls. Erk1/2 phosphorylation in M0, M1, and M2 macrophages was evaluated via immunoblotting (left). Quantification of p−Erk1/2 level in M1 macrophages was normalized to Erk1/2 (right). Values are means ± SD. * *p* < 0.05. (**B**) BMDMs were polarized into M1 macrophages in a time-dependent manner (4 h, 8 h, and 16 h). Immunoblotting was employed to analyze the phosphorylation of CaMKII (p-CaMKII) (left). Quantification of p−CaMKII level in M1 macrophages was normalized to CaMKII (right). Values are means ± SD. * *p* < 0.05. (**C**) The activity of the transcription factors CREB/ATF−1, NF−κB (p65), and STAT1 in M0, M1, and M2 macrophages were assessed by immunoblotting. Quantification of p-CREB and p-ATF1 levels in M1 macrophages were normalized to actin loading control. Values are means ± SD. * *p* < 0.05, ns, not significant. All data are representative of five independent experiments.

**Figure 3 cells-13-00322-f003:**
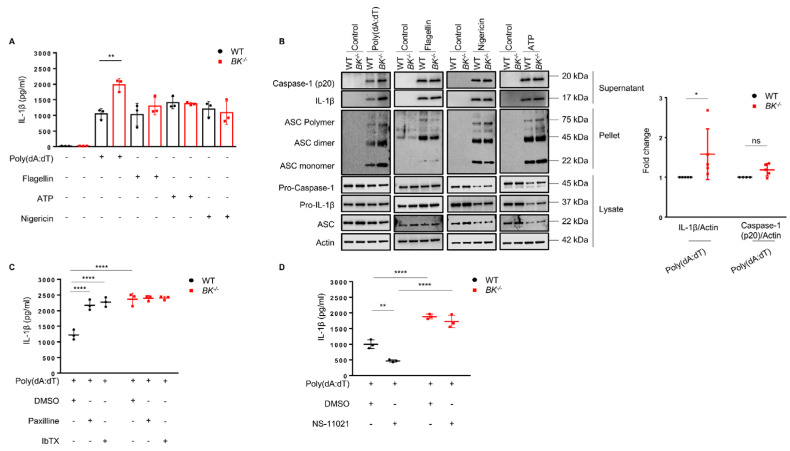
Inhibitory role of the BK channel on AIM2 inflammasome activation in macrophages. BMDMs were primed with LPS (100 ng/mL) for 4 h and subsequently activated with specific triggers of various inflammasomes. Macrophages were exposed to Poly(dA:dT) (1 µg/mL) for 50 min to induce AIM2 inflammasome activation, flagellin (0.5 µg/mL) for 1 h to induce NLRC4 inflammasome activation, and ATP (5 mM) for 45 min or nigericin (20 µM) for 30 min to induce NLRP3 inflammasome activation. (**A**) Cell culture supernatants were analyzed via ELISA to measure the release of IL−1β. Values are means ± SD. ** *p* < 0.01. Each graph represents data from three independent experiments. (**B**) Cell culture supernatants, pellets from cell lysates, and cell lysates were collected and analyzed for caspase−1 cleavage, IL−1β maturation, and ASC oligomerization. Quantification of IL−1β and caspase−1 (p20) levels in cell culture supernatant are normalized to actin. Values are means ± SD. * *p* < 0.05, ns, not significant. Each blot represents data from at least three independent experiments. (**C**) LPS-primed BMDMs were treated with or without BK channel inhibitors, paxilline (2.5 µM) or IbTX (100 nM), or DMSO for 30 min before inducing AIM2 inflammasome activation. IL−1β release was analyzed via ELISA in cell culture supernatants of activated macrophages. Values are means ± SD. **** *p* < 0.0001; Each graph represents data from three independent experiments. (**D**) LPS-primed BMDMs were treated with or without the BK channel activator, NS-11021 (2.5 µM), or DMSO for 30 min before inducing AIM2 inflammasome activation. IL−1β release was analyzed via ELISA in cell culture supernatants of activated macrophages. Values are means ± SD. ** *p* < 0.01, **** *p* < 0.0001. Each graph represents data from three independent experiments.

**Figure 4 cells-13-00322-f004:**
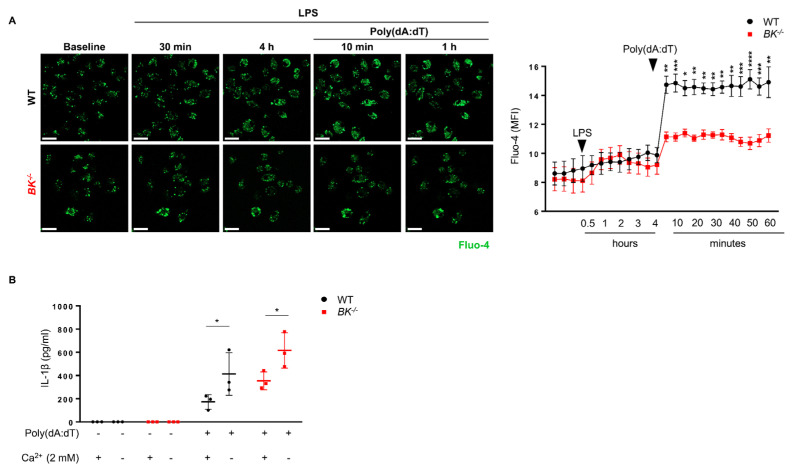
The absence of the BK channel limits extracellular Ca^2+^ influx while promoting IL-1β production following activation of the AIM2 inflammasome in macrophages. (**A**) BMDMs were labeled with Fluo−4 AM to monitor [Ca^2+^]_i_ levels using confocal microscopy. Representative immunofluorescence images (left) along with quantification of fluorescence intensity (right). Scale bars, 20 μm. Values are means ± SD. * *p* < 0.05, ** *p* < 0.01, *** *p* < 0.001, **** *p* < 0.0001; (**B**) LPS−primed BMDMs were stimulated with 1 µg/mL Poly(dA:dT) for AIM2 inflammasome activation in a solution with or without 2 mM Ca^2+^. The macrophage supernatants were analyzed after 1 h of stimulation, and the release of IL−1β was measured by ELISA. Values are means ± SD. * *p* < 0.05. Each graph represents data from three independent experiments.

**Figure 5 cells-13-00322-f005:**
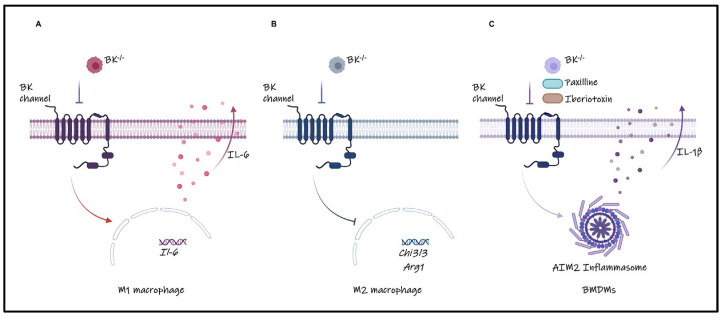
Graphical summary of the BK channel function in macrophages. The BK channels mediate macrophage polarization, acting as a determinant that curtails the secretion of pro−inflammatory cytokines such as IL−1β and IL−6 in macrophages. (**A**,**B**) Impairment or absence of the BK channel accentuates the propensity for M1 macrophage polarization while concurrently impeding the shift towards M2 polarization. (**C**) In bone marrow-derived macrophages (BMDMs), the inhibition or abrogation of the BK channel serves to facilitate AIM2 inflammasome activation, consequently amplifying the release of IL−1β. Created with BioRender.com.

**Table 1 cells-13-00322-t001:** Sequences of primer for qPCR assay.

Primer	Sequences
mouse *Nos2*	Forward: 5′-GTTCTCAGCCCAACAATACAAGA-3′
	Reserve: 5′-GTGGACGGGTCGATGTCAC-3′
mouse *Il6*	Forward: 5′-AGTGGTATAGACAGGTCTGTTGG-3′
	Reserve: 5′-CTGCAAGAGACTTCCATCCAG-3′
mouse *Il1b*	Forward: 5′-TGTAATGAAAGACGGCACACC-3′
	Reserve: 5′-TCTTCTTTGGGTATTGCTTGG-3′
mouse *Tnf*	Forward: 5′-CAGGCGGTGCCTATGTCTC-3′
	Reserve: 5′-CGATCACCCCGAAGTTCAGTAG-3′
mouse *Arg1*	Forward: 5′-CTCCAAGCAAAGTCCTTAGAG-3′
	Reserve: 5′-AGGAGCTGTCATTAGGGACATC-3′
mouse *Chi3l3*	Forward: 5′-CAGGTCTGGCAATTCTTCTGAA-3′
	Reserve: 5′-GTCTTGCTCATGTGTGTAAGTGA-3′
mouse *Eps8l1*	Forward: 5′-GTCTATCCTCACTCTAGCACTACC-3′
	Reserve: 5′-CCAACCAGCAGAAGCAGTAAG-3′
mouse *Kcnma1*	Forward: 5′-GACGCCTCTTCATGGTCTTC-3′
	Reserve: 5′-TAGGAGCCCCCGTATTTCTT-3′
mouse *Actb*	Forward: 5′-CACTGTCGAGTCGCGTCC-3′
	Reserve: 5′-TCATCCATGGCGAACTGGTG-3′

## Data Availability

The raw data are available upon request.

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
