# Peer review of "The BK Channel Limits the Pro-Inflammatory Activity of Macrophages"

_cells, 2024, doi:10.3390/cells13040322_

Round 1

Reviewer 1 Report

Comments and Suggestions for Authors

The manuscript entitled “The BK channel limits the pro-inflammatory activity of macro- 2 phages” compelling evidence for a role of the BK channel in macrophages in the regulation of their pro-inflammatory responses. The absence of the BK channel in macrophages results in increased M1 macrophage polarization, IL-6 secretion and AIM2 inflammasome activation. Furthermore, our investigation has revealed the involvement of Erk1/2 and CaMKII kinases, as well as the ATF-1 transcription factor, which are associated with BK channel activity. The manuscript needs minor revision before it can be accepted for publication.

1.     The author needs to include a few recent references relevant to the work.

2.     Improve the resolution of figures so that readers can better understand figure markings.

3.     It will be great if the authors can include a cartoon representation of conclusion.

4.     The authors need to read the MS again and make the sentences a little more grammatically correct.

Comments on the Quality of English Language

Minor editing needed

Reviewer 2 Report

Comments and Suggestions for Authors

In this paper, the authors study the role of BK channels (large-conductance voltage and calcium activated potassium channel) in the regulation of their pro-inflammatory responses of macrophages. Flow cytometry and western blot techniques demonstrate that the absence of BK channels results in increased M1 polarization, IL-6 secretion, and AIM2 inflammasome activation. Also, these channels seem to be related to Erk1/2 and CaMKII kinases, and the ATF-1 transcription factor.

Specific comments:

-         Authors should provide their data as scatter plots instead of bars. In this way, the reviewers and the readers can notice not only the number of experiments performed (which avoids the fact of reading the figure legend to know it and its lack in some cases such as Fig 3A and C) but also it possible to realize about the dispersion of the data.

-         I missed some information about the selection of the M1 and M2 cells based on canonical markers such as CD80/CD86 and CD206, respectively among others. Do the authors make any positive selection after their treatments to be sure of the nature of the populations they are working with?

-         Figure 1A should be provided in a similar resolution to other blots such as that shown in Fig. 1B

-         All the conclusions derived from blots (Figs. 1B, Fig 2, 3B) and flow cytometry data (Suppl Fig. 3C) have to be supported by a quantification. Especially in those cases were are not visually clear such as total levels of proteins or p-protein/total protein ratios. Please, show the quantifications.

-         In Fig. 2B, the authors show the levels of p-CAMKII through the time instead of M0, M1, and M2 polarization, however, some details about the reason underneath should be provided for a better clarification. At the same time, What is the meaning of the apparent increase and decrease in time? How are their levels in the different polarizations?

-         I miss more specific information about those experiments performed with paxillin, iberiotoxin, and NS-11021. Did the authors add firstly the inhibitors/activators, before the LPS-mediated activation? At the same time? After LPS? These details would determine the potential use of these drugs as a prevention or treatment, respectively. Considering this, What is the effect of the opposite order in these data? Is the LPS-mediated effect dependent on the basal BK channels activity or its effect can be modulated by the BK inhibition/activation later?

-         What is the meaning of the BK channels in the experiments shown in Fig. 4A. Due to the fact the BK channels are calcium-dependent channels, one would expect that the effect before of these channels, such as the calcium dynamics, should be unaffected however these data demonstrate that the absence of BK channels contribute in the calcium levels. How do you interpret this?

-         It has been previously demonstrated (PMID 33713314) that nuclear BK channels regulate CREB phosphorylation in Raw cells. What is the contribution of this microdomain to the results shown in this paper?

Minor points:

-         Please, show the sequences of the primer used in a table instead of within the text for a better clarification to the readers.

-         In Fig. 2C, p65 should be changed by NF-kB or at least added for better clarification to non-specialized readers. Alternatively, NF-kB should be changed by p65 in the text.

-     Include the meaning of ‘***’ in the figure legend 3D

Reviewer 3 Report

Comments and Suggestions for Authors

The authors of the manuscript titled 'The BK channel limits the pro-inflammatory activity of macrophages' attempt to investigate the previously unexplored role of KCa1.1 or BK channels in immune regulation and its potential role in modulating the AIM2 inflammasome activity. The authors have presented results from systematically designed experiments and verified that the BK channel plays an important role in regulating the influx of extracellular calcium in macrophages and limits AIM2 inflammasome activation. Overall, the manuscript is well-written, organized, and easy to interpret. The science presented within the manuscript is of the highest quality and the gap in the field addressed is of importance to further the field. The authors are advised the review the minor concerns raised below to further improve the quality of the presentation and make the manuscript useful for the reader.

1. All the western blot images should be carefully evaluated. Some images seem to be heavily contrasted. Example: Figure 1A is washed out. Could the authors present a better image since the experiment was repeated multiple times? 

2. The authors need to provide control blots for every individual protein tested. If the authors have used the same blot to probe different proteins it should be pointed out both in the methods section and the figure legend. Also, most western blot images seem to be cropped extensively. The authors are advised to provide images with more generous cropping highlighting molecular weight marker regions above and below the protein of interest. 

3. The catalog numbers for the antibodies and the kits used should be provided in the. methods section.

4. The authors are advised to elaborate more on the image acquisition parameters for calcium imaging and also provide more details on the quantification aspect where the authors should clearly outline the analysis steps.

5. The authors are advised to make a model figure that highlights and summarizes the findings from the paper.  

Reviewer 4 Report

Comments and Suggestions for Authors

This manuscript describes studies examining the role played by BKCa channels in regulation of the function of bone-marrow-derived macrophages, in vitro.  My major concern in trying to understand how loss of BKCa channel function depresses Ca2+ signaling but augments AIM2 inflammasome function.  While the authors have clearly demonstrated this relationship, they have not examined the cellular mechanisms responsible for this relationship.  This leads me to ask several questions that I would like the authors to try and answer.  First, what is the source of Ca2+ responsible for the putative activation of the BKCa channels in your macrophages?  Is it due to release of Ca2+ from internal stores via IP3R or is it due to Ca2+ influx across the plasma membrane? It is unlikely to be from Ca2+ through voyage-gated Calcium channels because one would expect loss of PBCa channel function to augment, not inhibit Ca2+ signaling.  Is the Ca2+ signal initiated by release of Ca2+ through IP3R and then supported by store-operated Ca2+ entry through STIM/ORAI or TRP channels? Second in the experiments shown in Figure 4A and 4B, after stimulation of WT cells, if BKCa channels are acutely inhibited with paxilline, what happens to the Ca2+ signal?  Does it go down as suggested by your findings with the BKCa-KO cells?  Third, what happens to the intracellular Ca2+ signal if extracellular Ca2+ is removed?  Please note that dependent on cell type, removal of extracellular Ca2+ not only eliminates Ca2+ influx, but also deplete intracellular Ca2+ stores.  I suspect that stimulation with Poly (dA:dT) causes release of Ca2+ from internal stores that activates BKCa channels.  This hyperpolarizes the plasma membrane increasing the electrochemical gradient for Ca2+ entry through non-voltage gated Ca2+ channels such as store-operated channels.  Loss of function of BKCa channels would eliminate the hyperpolarization and reduce the gradient for Ca2+ influx consistent with your data.  My final question is how does a decrease in Ca2+ signaling augment AIM2 inflammasome function????

Round 2

Reviewer 2 Report

Comments and Suggestions for Authors

Although the authors have explained in part some of the issues performed by me and the rest of the reviewers, for some reason they have not included any additional experiments to reinforce their conclusions. 

Author Response

We express our gratitude to the reviewer for his favorable response to our revised manuscript. His insightful comments and valuable suggestions have significantly enhanced the quality of our work, and we truly appreciate his constructive input.

In response to the reviewer's feedback, we would like to clarify that the revised manuscript has undergone substantial improvements in response to the feedback from all reviewers, including the issues highlighted by Reviewer 2. We have carefully considered each comment and implemented significant modifications to both the textual content and figures to enhance the overall quality of the manuscript.

While we appreciate the suggestion for additional experiments, Reviewer 2 did not specify any particular experiment to be performed or propose further modifications for the revised manuscript. Given the constraints of the limited revision time, we believe that we have adequately addressed the concerns align with the objectives of the study. Therefore, we hope that no further additional experiments are necessary at this stage.

Reviewer 4 Report

Comments and Suggestions for Authors

The authors have adequately addressed my concerns.  No additional comments.

Author Response

We express our gratitude to the reviewer for his favorable response to our revised manuscript.